# Clinical and growth outcomes after meconium-related ileus improved with Gastrografin enema in very low birth weight infants

**Woo Sun Song[1]°, Hye Sun Yoon[2]°, Seung Yeon Kim[2]\***

**1** Department of Medicine, Kyungpook National University, College of Medicine, Daegu, Korea,
**2** Department of Pediatrics, Nowon Eulji Medical Center, Eulji University School of Medicine, Seoul, South Korea

° These authors contributed equally to this work.
\* dunggiduk@eulji.ac.kr, dunggiduk5@gmail.com

## Abstract

### Background

Meconium-related ileus in very low birth weight infants can lead to increased morbidity or mortality and prolonged hospitalization without prompt diagnosis and treatment. This study primarily aimed to identify the incidence of and factors associated with meconium-related ileus and secondarily sought to investigate clinical and growth outcomes after water-soluble contrast media (Gastrografin) enema.

### Methods

We retrospectively reviewed medical records of very low birth weight infants born between February 2009 and March 2019 in the neonatal intensive care unit of a single medical center. Perinatal factors, clinical outcomes, and growth outcomes were compared between the group with meconium-related ileus that received Gastrografin enema and the control group.

### Results

Twenty-four (6.9%) patients were diagnosed with meconium-related ileus among 347 very low birth weight infants. All achieved successful evacuation of meconium with an average of 2.8 (range: 1–8) Gastrografin enema attempts without procedure-related complications. Initiation of Gastrografin enema was performed at mean 7.0 days (range: 2–16) after birth. Incidences of moderate to severe bronchopulmonary dysplasia were higher and the duration of mechanical ventilation and need for oxygen were longer in the meconium-related ileus group ($P = 0.039$, 0.046, 0.048, respectively). Meconium-related ileus infants took more time to start enteral feeding and the nothing per oral time was longer ($P = 0.001$ and 0.018, respectively). However, time to achieve full enteral feeding and Z-scores for weight and height at 37 weeks and at 6 months corrected age did not differ between the two groups.

**Data Availability Statement:** All relevant data are within the paper and its Supporting Information files.

**Funding:** The authors received no specific funding for this work.

**Competing interests:** The authors have declared that no competing interests exist.

## Conclusions

Gastrografin enema in very low birth weight infants with meconium-related ileus was an effective and safe medical management. Following Gastrografin enema, very low birth weight infants with meconium-related ileus achieved similar subsequent feeding progress and similar growth levels as the control groups without meconium-related ileus.

## Introduction

Meconium is a thick and tenacious material found in the bowel that develops in the fetal period and usually passes after birth [1]. Meconium is excreted within 48 h from birth in 99% of healthy full term infants; however, only 57% of preterm infants born under 29 weeks of gestational age can pass meconium spontaneously [2, 3]. Clatworthy et al. [4] first described meconium plug syndrome in 1956 that was caused by blockage of the distal colon by meconium and could not be passed spontaneously. Rickham and Boeckman [5] first described meconium disease in 1965 as the meconium plugs found in the distal ileum and proximal colon, which were not associated with cystic fibrosis and appeared more common in very low birth weight (VLBW, birth weight <1,500 g) infants. In 1999, Kubota et al. proposed that the term meconium-related ileus (MRI) should include meconium plug syndrome and meconium disease because both diseases have speculated similar pathogenesis [6]. The pathogenesis of MRI is suggested by immature or ineffective peristalsis of the fetal intestine combined with excessive water absorption [7, 8]. Failure to pass the meconium causes feeding intolerance, bilious vomiting, and progressive abdominal distension [9]. Delayed enteral nutrition can cause the intestinal membranes to atrophy and reduce the absorption rate that can destroy the barrier function of the intestinal mucosa and prevent absorption of nutrients by the intestine [10]. In addition, the risk of sepsis and cholestasis is increased due to the longer duration of parenteral nutrition by delaying full enteral feeding [11]. Eventually, this leads to extrauterine growth restrictions, and may have long-term adverse effects, including short stature and poor neurodevelopmental outcomes [12]. Reasons for delayed passage may include meconium ileus that is related to cystic fibrosis (CF), and meconium-related ileus (MRI) without CF, which occurs mainly in preterm infants [13]. Several studies have reported treatment of MRI with saline enemas, glycerin suppositories, and oral contrast agents [14]. Whether these methods are associated with the evacuation of the meconium, thus reducing the time to full enteral feeds in VLBW infants, remains controversial [11, 14, 15]. Other studies have suggested that water-soluble contrast media (Gastrografin; diatrizoate meglumine and diatrizoate sodium) enema is effective for meconium evacuation [16–20]. However, there are insufficient definite treatment guidelines and evidence of the effectiveness of treatment for VLBW infants with MRI. Furthermore, no studies about growth outcomes of VLBW infants after MRI excretion have been done. This study primarily aimed to identify the incidence of and factors associated with meconium-related ileus and secondarily sought to investigate clinical and growth outcomes after water-soluble contrast media (Gastrografin) enema.

## Materials and methods

### Study design and population

This study was a retrospective analysis of VLBW infants who were born from February 1, 2009 to March 1, 2019 at the neonatal intensive care unit (NICU) of Daejeon Eulji Medical Center.

Medical records of VLBW infants with MRI were reviewed for demographic characteristics, perinatal risk factors, patient clinical data, treatments, and outcomes. Patient clinical data included the Apgar score, small for gestational age (SGA, birthweights below the $10^{th}$ percentile) [21], magnesium concentration (within 24 h after birth), respiratory distress syndrome (RDS), and patent ductus arteriosus (PDA). Perinatal risk factors included antenatal steroid administration, preeclampsia, pregnancy-induced hypertension (PIH), gestational diabetes mellitus (GDM), premature rupture of membranes (PROM), multiple pregnancy, chorioamnionitis, oligohydramnios, polyhydramnios, maternal hypothyroidism, and maternal age. MRI was considered a possibility when the patients had a problem with passing meconium and showed progressive abdominal distention and feeding intolerance despite glycerin or saline enema. Unlike in NEC, in MRI there is persistent of progressive gaseous bowel distention without high air fluid levels or bowel wall edema noted on plain radiography, and patients presented with a relatively benign general condition and normal laboratory test results. GaE was performed as the next step for unresolved MRI after several attempts of routine procedures, such as rectal stimulation, glycerin enema, and warm saline enema. We excluded infants with chromosomal anomalies, other congenital disorders, and pre-existing gastrointestinal abnormalities. Based on the treated VLBW infants with MRI, a matched control group (one-to-two matching, comparable gestational age) without MRI in VLBW infants was formed. We investigated the following morbidities in the NICU: intraventricular hemorrhage (IVH) [22], retinopathy of prematurity (ROP) [23], bronchopulmonary dysplasia (BPD) [24], and late-onset sepsis (LOS). LOS was diagnosed when symptoms appeared after 72 h of birth and culture of blood bacteria. We also reviewed the duration of mechanical and non-invasive ventilation, duration of oxygen administration, total period of hospitalization, and mortality. We assessed the time to full enteral feeds, and the Z-scores of weight and height at the corrected ages of 37 weeks and 6 months to compare feeding progress and growth in the two groups. The time to reach full enteral feeding was defined as the time from birth to the time when a 120 ml/kg full feed was achieved, and parenteral nutrition was stopped over a period of 24 h. The Z-scores of weight and height at birth and at the corrected age of 37 weeks were defined according to the Fenton growth chart [25]. The Z-score at the corrected 6 months of age was defined according to the World Health Organization child growth standards [26]. The study protocol was reviewed and approved by the Institutional Review Board of Eulji University Hospital (No. 2019-06-032). The requirement for informed consent was waived due to the retrospective nature of this study.

## Protocol of enema with water-soluble contrast media, Gastrografin

We diluted water-soluble contrast media (Gastrografin®, 2150 mOsm/L; Bayer Healthcare, Newbury, England) with normal saline (286 mOsm/L) at a ratio of 1:3 (752 mOsm/L). We prepared 20 mL of mixture, using 5 mL of Gastrografin and 15 mL of normal saline. A 6 or 8-Fr Foley catheter with its tip positioned in the rectosigmoid colon was used and approximately 15–20 mL of the diluted Gastrografin with less 1 mL/sec velocity, was manually injected. The Foley catheter was not ballooned to reduce the risk associated with pressure. The practitioner performed the procedure while monitoring the clinical condition of the abdomen (i.e., inspection of distension and palpation of tension) without fluoroscopy or ultrasound guidance in the NICU. The amount of diluted Gastrografin injection was controlled in accordance with the clinical condition of the abdomen. If resistance or over distension of the abdomen were encountered, the injection was ceased. After Gastrografin injection, the practitioner obstructed the anus with gauze to minimize leakage and maximize the effectiveness of enema for approximately 5–10 min. Other health care providers performed portable plain radiography after

enema to identify the level reached by the injected Gastrografin. GaE was not repeated immediately, even if Gastrografin had not reached the appropriate position, which is considered the level of the ileocecal valve on the X-ray [18]. GaE was performed repeatedly in ineffective cases such as those in whom the meconium was not excreted within 24 h after GaE, or those with an increase in abdominal distension, on a once-per-day basis. All Gastrografin injections were performed by two neonatologists (SYK, WSS) and vital signs, such as heart rate, oxygen saturation, and blood pressure, were continuously monitored throughout the entire procedure. Follow-up abdominal plain radiography was done after 6–12 h. If symptoms of MRI were not relieved, GaE was performed daily until the resolution of MRI.

### Statistical analysis

SPSS version 22.0 (SPSS, Chicago, IL) was used for statistical analysis. Student's t-test, chi-square test, and Fisher's exact test were used to compare frequencies. A $P$ value of <0.05 was considered to be statistically significant.

## Results

### Study population and characteristics

A total of 347 VLBW infants were born during the study period. Among them, 24 (6.9%) infants with MRI were treated using GaE (Fig 1).

The mean gestational age was 28.6±2.8 weeks (range; 23 weeks 3 days–35 weeks 6 days) and the mean birth weight was 1,054.1±235.4 g (range; 650 g–1,450 g) in 24 VLBW infants with MRI. Compared to the control group, gestational age and birth weight were not statistically different.

### Perinatal risk factors and clinical outcomes

We did not find significant associations between MRI and any of the following: SGA, Mg concentration, RDS, and PDA (Table 1). There were no statistical differences between the MRI group and the control group in terms of perinatal risk factors, such as GDM, PIH, preeclampsia, perinatal steroid use, oligohydramnios, hypothyroidism of the mother, chorioamnionitis, and PROM (Table 1). The incidence of moderate to severe BPD was higher in VLBW infants with MRI ($P$ = 0.039). Furthermore, the durations of mechanical ventilation and oxygen use were longer in the patient group ($P$ = 0.048). There was no statistical difference in terms of mortality, hospitalization period and other morbidities, such as IVH (stage ≥3), ROP, or LOS (Table 2).

### Treatment of GaE

Twenty-four VLBW infants with MRI were treated with GaE. Symptoms of MRI developed at a mean of 5.7 days (range: 2–16 days) and GaE was performed at a mean of 7.0 (range: 2–16) days after birth. The mean frequency of GaE was 2.8 (range: 1–8) (Table 3). When GaE was performed, it was advanced to the distal ileum in four cases (No. 2, 8, 17, and 18); ascending colon in 17 cases (No. 1, 5, 6, 7, 9, 10, 11, 12, 13, 14, 15, 16, 19, 20, 21, 23, and 24); and transverse colon in three cases (No. 3, 4, and 22) (Table 3, Figs 1 and 2). On abdomen radiography, the time to resolution of MRI was a mean of 12.2 days (range: 5–21 days) after birth (Table 3). In other words, MRI resolved at a mean of 5.2 days after GaE. There were no complications, such as perforation or dehydration, related to the procedure during or after administering GaE.

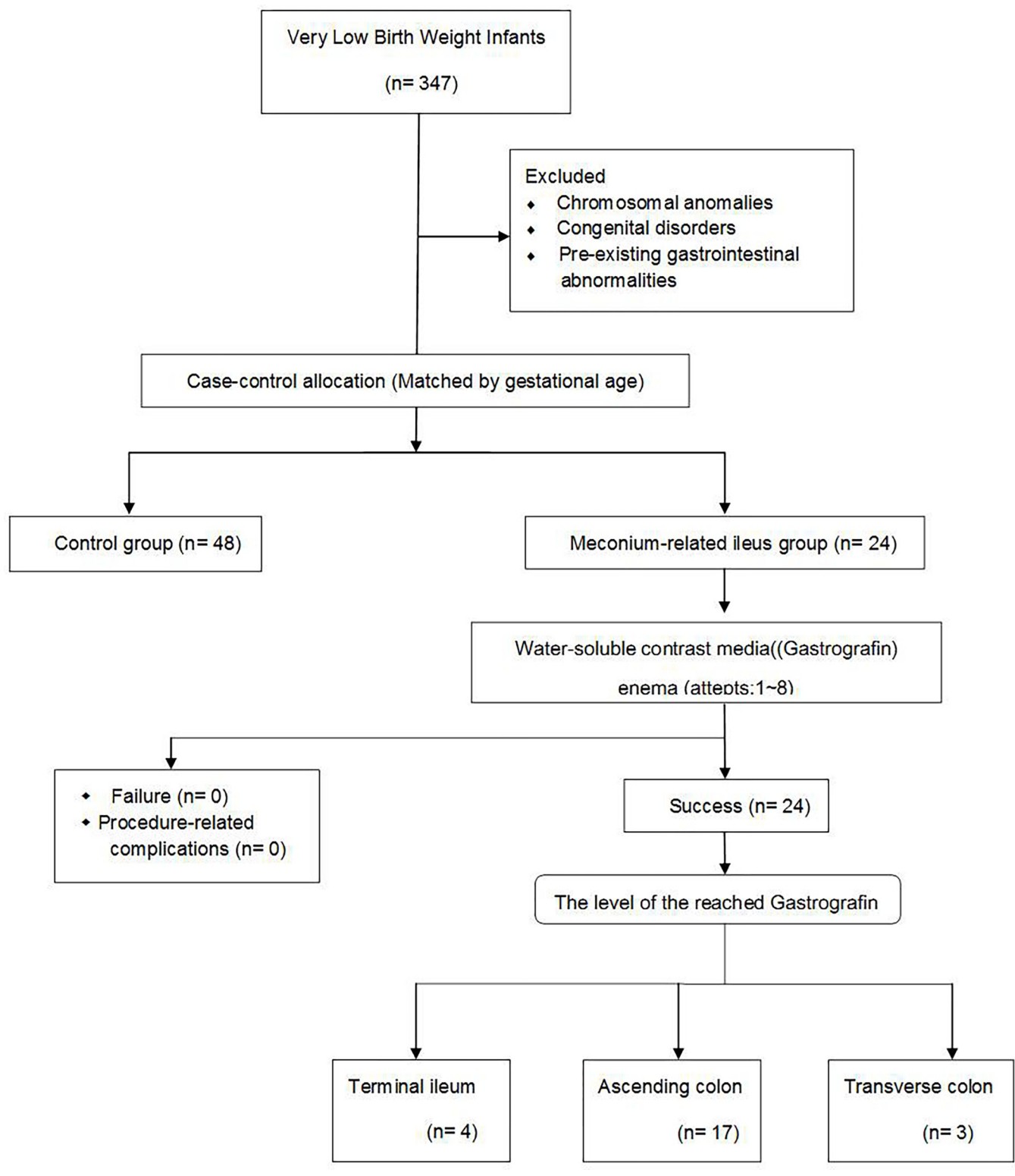

**Fig 1. Study diagram.**

**Table 1. Patients' perinatal history and demographics and clinical characteristics.**

| Characteristics | Case | Control | P value |
|---|---|---|---|
| | (n = 24) | (n = 48) | |
| Maternal age (SD, year) | 33.8±5.7 | 33.2±3.8 | 0.598 |
| Maternal preeclampsia, n (%) | 7/24 (29.2) | 14/48 (29.2) | 1.000 |
| Maternal PIH, n (%) | 7/24 (29.2) | 13/48 (27.1) | 0.852 |
| Maternal GDM, n (%) | 4/24 (16.7) | 6/48 (12.5) | 0.722 |
| Maternal hypothyroidism, n (%) | 2/24 (8.3) | 5/48 (10.4) | 1.000 |
| Multiple pregnancies, n (%) | 13/24 (54.2) | 21/48 (43.8) | 0.404 |
| Chorioamnionitis, n (%) | 2/24 (8.3) | 11/48 (22.9) | 0.196 |
| Antenatal steroid, n (%) | 17/24 (70.8) | 39/48 (81.3) | 0.316 |
| Oligohydramnios, n (%) | 2/24 (8.3) | 6/48 (12.5) | 0.710 |
| PROM, n (%) | 8/24 (33.3) | 12/48 (25.0) | 0.457 |
| Polyhydramnios, n (%) | 0/24 (0.0) | 0/48 (0.0) | - |
| Gestational age, mean (SD, weeks) | 28.6±2.8 | 28.6±2.7 | 0.953 |
| Birth weight, mean (SD, g) | 1054.1±235.4 | 1073.7±256.1 | 0.755 |
| Head circumference (SD, cm) | 25.6±2.4 | 25.5±2.0 | 0.969 |
| Height (SD, cm) | 36.4±3.0 | 36.8±2.9 | 0.595 |
| Male, n (%) | 11/24 (45.8) | 26/48 (54.2) | 0.505 |
| 1-min APGAR score, median (IQR) | 4 (1–7) | 4 (1–9) | 0.108 |
| 5-min APGAR score, median (IQR) | 6 (3–9) | 6 (2–9) | 0.408 |
| Small for gestational age (%) | 5/24 (20.8) | 8/48 (16.7) | 0.749 |
| Cesarean delivery, n (%) | 19/24 (79.2) | 40/48 (83.3) | 0.749 |
| Mg concentration, within 24hr (SD, mg/dL) | 2.4±0.7 | 2.5±1.1 | 0.773 |
| RDS (%) | 21/24 (87.5) | 42/48 (87.5) | 1.000 |
| PDA (%) | 20/24 (83.3) | 38/48 (79.2) | 0.761 |
| Pharmacological PDA treatment (%) | 13/24 (54.2) | 22/48 (45.8) | 0.505 |
| PDA ligation (%) | 8/24 (33.3) | 6/48 (12.5) | 0.056 |

Abbreviations: PIH, pregnancy induced hypertension; GDM, gestational diabetes mellitus; PROM, premature rupture of membranes; RDS, respiratory distress syndrome; PDA, patent ductus arteriosus.

After GaE treatment, two patients died. These deaths were not associated with Gastrografin complications. After improvement of MRI, the death of patient 2 occurred due to renal failure at 129 days, and patient 6 due to late onset sepsis by methicillin resistant *Staphylococcus aureus*. Patient 21 developed perforation at the proximal ileum four days after the MRI resolved, not during GaE; there was improvement after recovery operation and the patient was discharged 43 days after birth.

## Feeding progress and growth

The comparison of feeding progress and growth between cases and controls showed that the duration of nothing per oral (NPO) was longer (*P* = 0.018) and the first enteral feeding day was much later (*P* = 0.001) in VLBW infants with MRI. However, there was no difference in the number of days taken to achieve full enteral feed or the duration of parenteral nutrition. The Z-scores of weight and height at birth, and corrected age at 37 weeks and 6 months were not statistically different (Table 4).

## Discussion

Although MRI is a known multifactorial disease developed in VLBW infants with immature gastroduodenal and colonic motility, which may also contribute to feeding intolerance and

**Table 2. Comparison of clinical outcomes between case and control group.**

| Morbidities and clinical outcomes | Case | Control | Odds ratio | P value |
|---|---|---|---|---|
| | (n = 24) | (n = 48) | (95% confidence interval) | |
| Severe IVH, stage ≥3 (%) | 1/24 (4.2) | 2/48 (4.2) | 1.00 (0.086–11.613) | 1.000 |
| ROP stage ≥ 2 (%) | 5/24 (20.8) | 13/48 (27.1) | 0.709 (0.219–2.289) | 0.564 |
| ROP laser (%) | 5/24 (20.8) | 12/48 (25.0) | 0.789 (0.242–2.575) | 0.695 |
| Moderate to severe BPD (%) | 13/24 (54.2) | 14/48 (29.2) | 2.87 (1.039–7.927) | 0.039* |
| Mechanical ventilation | 21/24 (87.5) | 42/48 (87.5) | 1.000 (0.227–4.400) | 1.000 |
| Duration of Mechanical ventilation (SD, day) | 27.0±37.2 | 13.2±20.8 | | 0.046* |
| Duration of CPAP (SD, day) | 11.9±14.2 | 12.2±14.2 | | 0.940 |
| Duration of Nasal High Flow Oxygen (SD, day) | 28.8±23.8 | 24.4±15.0 | | 0.346 |
| Duration of Oxygen (SD, day) | 70.1±54.6 | 50.1±29.8 | | 0.048* |
| Hospitalization period (SD, day) | 97.9±51.2 | 85.7±29.8 | | 0.211 |
| Late onset sepsis (%) | 3/24 (12.5) | 3/48 (6.3) | 2.143 (0.399–11.521) | 0.393 |
| Mortality (%) | 2/24 (8.3) | 1/48 (2.1) | 4.273 (0.368–49.676) | 0.256 |

Abbreviation: IVH, Intraventricular hemorrhage; ROP, retinopathy of prematurity; BPD, bronchopulmonary dysplasia; CPAP, continuous positive airway pressure
*Statistically significant (P<0.05)

**Table 3. Clinical course of VLBWI with MRI receiving Gastrografin enema.**

| Case No. | Sex | GA | BW (g) | MRI appeared (day of age) | Gastrografin enema administered (days of age) | The level reached by Gastrografin | MRI resolved (days of age) | Duration of NPO (day) | Full enteral feeding day (120cc/day) | Prognosis | Age at discharge (days) |
|---|---|---|---|---|---|---|---|---|---|---|---|
| 1 | M | 23w5d | 730 | 16 | 16,17 | A,A | 19 | 26 | 53 | S | 175 |
| 2 | F | 24w3d | 650 | 7 | 9 | DI | 11 | 65 | N | Expire | 129 |
| 3 | M | 26w1d | 990 | 14 | 15,17 | D,T | 18 | 5 | 29 | S | 109 |
| 4 | F | 26w1d | 900 | 13 | 14,17 | T,T | 18 | 5 | 34 | S | 102 |
| 5 | M | 26w3d | 830 | 7 | 7 | A | 12 | 21 | 116 | S | 187 |
| 6 | M | 27w0d | 1100 | 6 | 6,7, 8 | D,A,A | 10 | 21 | N | Expire | 33 |
| 7 | M | 27w2d | 1170 | 3 | 4,5,6 | D,T,A | 8 | 26 | 106 | S | 156 |
| 8 | F | 27w2d | 580 | 2 | 2,7,8,9,10,12,13,16 | D,T,A,A,A,A,DI,A, | 17 | 13 | 44 | S | 241 |
| 9 | F | 27w3d | 990 | 6 | 9,10 | A,A | 13 | 6 | 27 | S | 108 |
| 10 | F | 27w4d | 1120 | 2 | 2,4 | T,A | 7 | 13 | 32 | S | 51 |
| 11 | F | 27w5d | 1110 | 2 | 7 | A | 10 | 9 | 36 | S | 105 |
| 12 | F | 28w1d | 980 | 12 | 12,13,14 | T,T,A | 16 | 9 | 64 | S | 103 |
| 13 | M | 28w2d | 870 | 4 | 6 | A | 8 | 13 | 60 | S | 67 |
| 14 | M | 28w4d | 880 | 2 | 5,6,7,8 | D,D,A,A | 9 | 13 | 28 | S | 110 |
| 15 | F | 28w5d | 1230 | 1 | 2,3,4 | A,A,A | 7 | 5 | 41 | S | 83 |
| 16 | M | 29w2d | 1370 | 2 | 4,5,6,7,8 | D,T,T,T,A | 11 | 2 | 24 | S | 61 |
| 17 | F | 29w3d | 1130 | 8 | 8 | DI | 10 | 5 | 23 | S | 63 |
| 18 | F | 29w4d | 1000 | 2 | 2,3,8,10,11,12,13 | D,DI,A,DI,A,T,T | 15 | 1 | 22 | S | 64 |
| 19 | F | 29w6d | 1210 | 16 | 16 | A | 18 | 12 | 34 | S | 72 |
| 20 | M | 30w1d | 970 | 1 | 4,5,7 | T,A,A | 9 | 10 | 70 | S | 98 |
| 21 | M | 30w3d | 1350 | 1 | 4,5 | A,A | 5 | 17 | 28 | S | 43 |
| 22 | F | 33w2d | 1460 | 1 | 3,4,6 | D,D,T | 7 | 8 | 23 | S | 58 |
| 23 | M | 33w6d | 1240 | 1 | 3,4,6 | D,D,A | 9 | 3 | 16 | S | 38 |
| 24 | F | 35w6d | 1440 | 10 | 10,11,13,16,20 | D,D,D,A,T | 22 | 10 | 36 | S | 55 |

Abbreviation: GA, gestational age; BW, birth weight; MRI, meconium-related ileus; NPO, nothing per oral; M, male; F, female; A, ascending colon; T, transverse colon; D, descending colon; DI, distal ileum; S, success; N, not to reached full entering feeding.

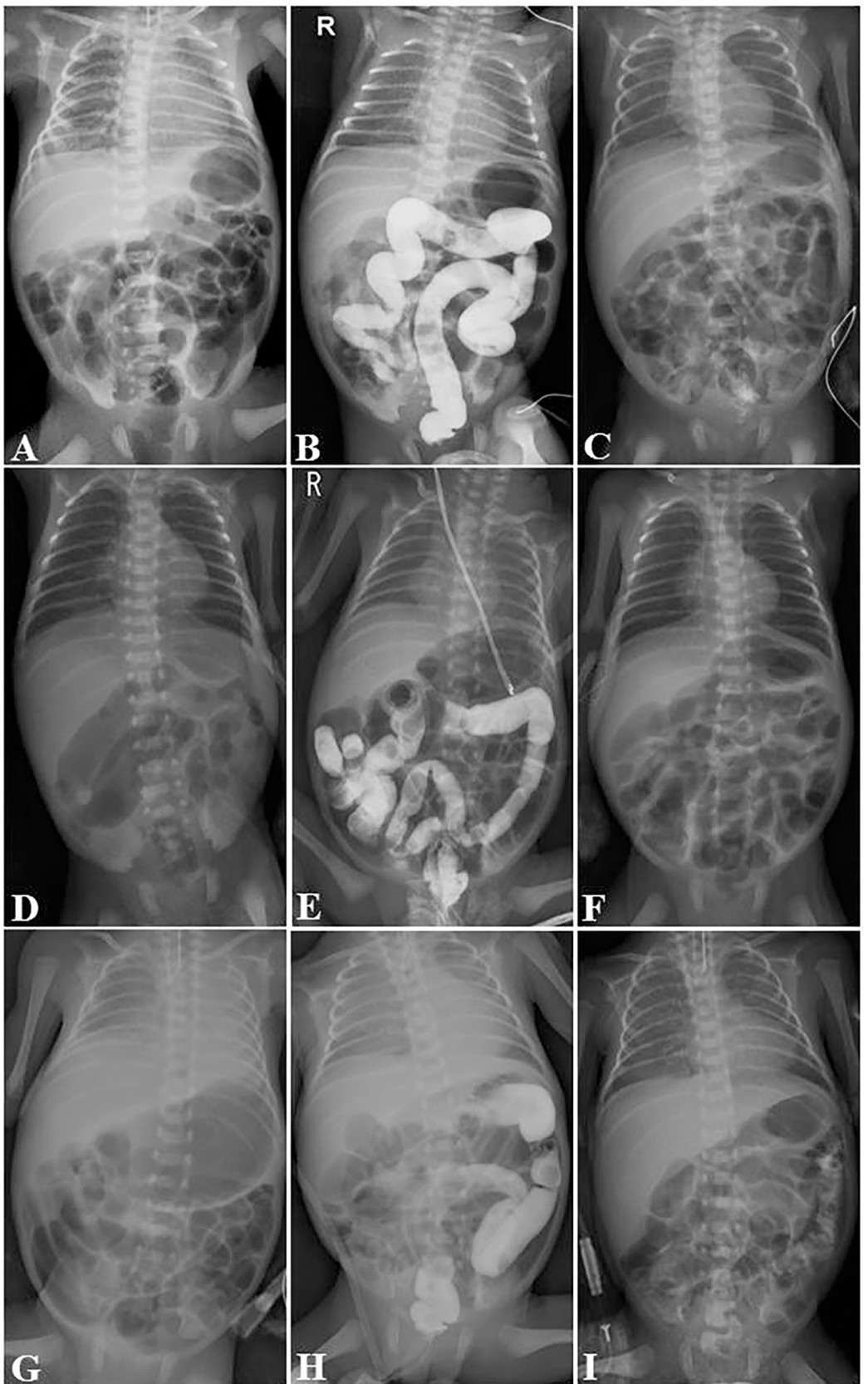

**Fig 2. Abdominal radiographs.** Abdominal radiograph of patient No. 18 (A, B, C), 20 (D, E, F), and 22 (G, H, I) showing dilated intestine before Gastrografin administration (A, D, E). Gastrografin was advanced to distal ileum (2nd try, B), ascending colon (3rd try, E), and transverse colon (3rd try, H) after Gastrografin enema. Abdominal radiograph C, F, and I show that intestinal ileus has been resolved due to meconium obstruction.

delayed meconium passage [27], the pathogenesis is still unclear. With advances in neonatal care, the number of MRI cases has increased along with VLBW infants' survival rates. Recent studies have reported MRI incidence at 3.9–11.3% in VLBW infants [8, 17, 28]. Consistent with this, in the current study, MRI occurred in 6.9% of VLBW infants. Perinatal risk factors associated with MRI are known maternal diabetes mellitus, PIH, use of $MgSO_4$ in the perinatal period, cesarean delivery, PROM, and twin pregnancies [8, 19, 29, 30]. These factors have been reported to be associated with intestinal hypoperfusion and intestinal dysmotility [8, 29, 31, 32]. In contrast to these risk factors, maternal steroid treatment was suggested to be protective MRI [30, 33]. In the current study, all of the previously suggested perinatal associated factors of MRI showed no statistically significant difference between the MRI and control groups. However, incidences of maternal GDM, maternal PIH, PROM, multiple pregnancies, and cesarean delivery in the MRI group resulted in 16.7%, 29.2%, 33.3%, 54.2%, and 79.2%, respectively. These results are consistent with those of previous studies [8, 20, 31] and suggest increased awareness and concern for a patient with perinatal risk factors, even if they were not significantly statistically different from the control group in the current study.

The retained meconium, often resulting in intestinal obstruction in VLBW infants, may cause significant morbidity and even mortality [34]. MRI is known to be associated with severe prematurity and low birth weight causing intestinal dysmotility and meconium obstruction [19, 29, 34]. Interestingly, the results of development of moderate to severe BPD, duration of mechanical ventilation, and duration of oxygen, which are associated with severe prematurity and low birth weight, were significantly different in the MRI group than in the control group in the current study. We could not find any reports on the association between respiratory dysfunction and MRI. Hence, we speculate that immature respiratory neuromuscular function

**Table 4. Comparison of growth and feeding progress between case and control group.**

| Outcome | Case | Control | P value |
|---|---|---|---|
| | (n = 24) | (n = 48) | |
| Start of enteral feeding (SD, day) | 3.43±2.3 | 1.4±1.0 | ≤0.001* |
| Full enteral feeding time, 120 ml /kg (SD, day) | 43.0±26.2 | 34.0±20.6 | 0.128 |
| Initial meconium pass day (SD) | 1.4±0.9 | 1.6±1.5 | 0.628 |
| Duration of NPO (SD, day) | 13.2±13.0 | 6.0±11.1 | 0.018* |
| Duration of PN (SD, day) | 55.9±32.2 | 43.0±26.1 | 0.074 |
| Z-score at birth | | | |
| Weight | -0.30±1.07 | -0.25±1.06 | 0.846 |
| Height | 0.03±1.00 | 0.03±1.27 | 0.990 |
| Z-score at CA 37 weeks | | | |
| Weight | -1.88±1.26 | -1.42±1.28 | 0.169 |
| Height | -1.77±1.23 | -1.32±1.24 | 0.169 |
| Z-score at CA 6 months | | | |
| Weight | -0.30±1.56 | -0.12±1.13 | 0.608 |
| Height | -0.33±2.17 | 0.06±1.36 | 0.373 |

Abbreviation: NPO, nothing per oral; PN, parenteral nutrition; CA, corrected age; *Statistically significant ($P<0.05$)

may be the cause since immature intestinal neuromuscular function has been proposed as one of the mechanisms of MRI; however, more research is needed. In contrast, the incidence of severe IVH, ROP, late onset sepsis, hospitalization period, and mortality investigated as other clinical outcomes in the MRI group were not statistically significantly different than those in the control group investigated in the current study. However, the relationship between those clinical outcomes and MRI needs further study.

In 1969, Noblett [35] first described the practice of water-soluble contrast agents, especially Gastrografin, and since then, GaE has been considered a non-invasive and effective medical treatment for MRI with a success rate of ≥70% [31, 36–39]. Fluoroscopy is used to confirm that the distal ileum has been reached when GaE is performed on MRI patients. Recently, ultrasound-guided enema, which can be performed at bedside in the NICU, thereby avoiding transport of the infant to the fluoroscopic room, has been used to maintain body temperature and humidity, and ensure minimal handling of patients [28, 36]. In the current study, all GaE procedures were performed by two neonatologists in NICU without ultrasound or fluoroscopy and we confirmed the reached point of Gastrografin by portable X-ray after GaE. Many previous studies recommended that the Gastrografin reach above the level of the ileocecal valve in MRI [16–19, 28, 31, 35]. Though Gastrografin did not reach the distal ileum in all cases, the results were successful. In 21 cases, Gastrografin reached the distal ileum and the right colon, and in the 3 remaining cases, it reached the transverse colon. The reason for successful meconium evacuation, despite Gastrografin not reaching the distal ileum, in the current study is not exactly known. It is highly probable that the early initiation of therapy and the frequent interventions, which were reported in previous studies [13, 17, 19, 29, 40], played important roles; however, it is also possible that our cases had a mild form of MRI. Emil et al. [29] indicated that a 10-day duration of obstruction was the limit for which the medical treatment was effective. In our study, GaE was performed on an average of 7.0 days after birth, with a duration of obstruction of < 10-days. In addition, repeated enemas were performed in 18 cases and the mean frequency of GaE was 2.8 times. We also speculate that our procedure, which involved obstructing the anus with gauze for approximately 5–10 min after Gastrografin injection, may have enhanced the interstinal peristalsis due to Gastrografin [11, 28]. Besides, there were no procedure-related complications such as perforation or dehydration in current study unlike previous studies [8, 20, 28, 41]. Based on the results of our study, we cautiously suggest that even if the location of the contrast agent is not the distal ileum, performing the procedure early and frequently while carefully observing the clinical findings of the patient by a skilled neonatologist can successfully promote meconium excretion as well as reduce procedure-related complications.

MRI is known to result in delaying full enteral feeding and subsequently induce poorer short- and long-term outcomes in VLBW infants [42]. In the current study, the first enteral feeding day was significantly delayed and the duration of NPO was significantly longer in the MRI group than in the control group because symptoms of ileus such as feeding intolerance, bilious vomiting, and abdominal distension were observed. However, the time to achieve full enteral feeding did not differ significantly in the two groups. This suggests that appropriate evacuation of meconium improves feeding intolerance in MRI patients. Furthermore, no significant difference in growth outcomes in the two groups at the corrected ages of 37 weeks and 6 months was observed. Based on these results, we suggest that meconium evacuation by GaE is significantly helpful for growth in VLBW infants with MRI. To the best of our knowledge, the current study is the first to report growth outcomes after resolved MRI by GaE treatment.

This study has several limitations. First, this was a retrospective study. Second, as the sample size in the study was small, bias may exist. Third, this study included subjects from only one center. Despite these limitations, this study is clinically meaningful; it provides the basis for the

usefulness of GaE as a treatment for MRI in VLBW infants, and is also the first study to report the growth outcomes of MRI patients after MRI resolution by GaE in VLBW infants. We suggest further studies with a large number of cases to confirm our findings.

## Conclusions

GaE is an effective and safe treatment for VLBW infants with MRI for evacuation of meconium. Furthermore, VLBW infants with MRI who receive GaE have a similar prognosis in terms of subsequent feeding success as the control group. Lastly, growth at 37 weeks and 6 months corrected ages did not differ significantly from those of control infants.

## Supporting information

**S1 File.**
(SPV)

**S1 Data.**
(XLSX)

**S2 Data.**
(SAV)

## Author Contributions

**Conceptualization:** Woo Sun Song, Hye Sun Yoon, Seung Yeon Kim.

**Data curation:** Woo Sun Song, Hye Sun Yoon, Seung Yeon Kim.

**Formal analysis:** Woo Sun Song, Hye Sun Yoon, Seung Yeon Kim.

**Investigation:** Woo Sun Song, Hye Sun Yoon, Seung Yeon Kim.

**Methodology:** Woo Sun Song, Seung Yeon Kim.

**Project administration:** Woo Sun Song, Hye Sun Yoon, Seung Yeon Kim.

**Resources:** Woo Sun Song, Hye Sun Yoon, Seung Yeon Kim.

**Software:** Woo Sun Song.

**Supervision:** Hye Sun Yoon, Seung Yeon Kim.

**Validation:** Woo Sun Song, Hye Sun Yoon, Seung Yeon Kim.

**Visualization:** Woo Sun Song, Hye Sun Yoon, Seung Yeon Kim.

**Writing – original draft:** Woo Sun Song, Seung Yeon Kim.

**Writing – review & editing:** Hye Sun Yoon, Seung Yeon Kim.

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
