## [Decision Letter · Decision Letter 0]

11 May 2022

PONE-D-22-07282Clinical and growth outcomes after meconium-related ileus improved with Gastrografin enema in very low birth weight infants.PLOS ONE

Dear Dr. Kim,

Thank you for submitting your manuscript to PLOS ONE. After careful consideration, we feel that it has merit but does not fully meet PLOS ONE’s publication criteria as it currently stands. Therefore, we invite you to submit a revised version of the manuscript that addresses the points raised during the review process.

We look forward to receiving your revised manuscript.

Kind regards,

Prem Singh Shekhawat, MD

Academic Editor

PLOS ONE

Journal Requirements:

4. Please ensure that you refer to Figure 2 in your text as, if accepted, production will need this reference to link the reader to the figure.

Additional Editor Comments:

Thank you for your submission titled “Clinical and growth outcomes after meconium-related ileus improved with Gastrografin enema in very low birth weight infants.” for publication in PLOA One. Kindly find attached comments by the two reviewers. Kindly modify your submission as suggested and resubmit for consideration to publish if you so desire. Thanks

Reviewers' comments:

Reviewer's Responses to Questions

**Comments to the Author**

1. Is the manuscript technically sound, and do the data support the conclusions?

Reviewer #1: Yes

Reviewer #2: Yes

2. Has the statistical analysis been performed appropriately and rigorously? 

Reviewer #1: Yes

Reviewer #2: Yes

3. Have the authors made all data underlying the findings in their manuscript fully available?

Reviewer #1: Yes

Reviewer #2: Yes

4. Is the manuscript presented in an intelligible fashion and written in standard English?

Reviewer #1: Yes

Reviewer #2: No

5. Review Comments to the Author

Reviewer #1: The authors reported the incidence, presentation, and likely associated prenatal risk factors with meconium-related ileus in very low birth weight (VLBW) newborns and investigated the clinical and the growth outcomes after gastrographin enema of 24 infants. All infants had successful evacuation without procedure-related complications. Despite the later and longer time to start enteral feeding in the study group compared to the control group; the time to reach full enteral feeding was statistically insignificant. The treatment group required longer respiratory support and had a higher incidence of moderate to severe bronchopulmonary dysplasia. The novelty of the study is reporting the Z-scores for weight and height for VLBW infants with meconium-related ileus at term corrected gestational age and 6 months corrected age which was statistically insignificant between the study group and the control group. Despite the small sample size of this study, it confirms the importance of early identification of meconium-related ileus, the efficacy of gastrographin enema treatment after trial of glycerin/saline enema, and that gastrographin enema is a safe option for these patients.

Reviewer #2: - Please concise the previous studies related to meconium related ileus if possible in the introduction section.

- We define the primary aim and secondary aims more clearly in the hypothesis section.

- Can we combine the table 1 and 2. Keep the maternal factors first followed by infant factors.

- Can we look into the factors why there is delay in start of the feeds after in gastrograffin group?

- I would advised to keep the hypothesis focused to the characteristics of MRI and immediate outcomes

- I would take the growth outcomes out of the hypothesis.

- Can we change the MRI to meconium associated ileus.

- The BPD effect is more related to gestation age rather than MRI. I would not include that in the conclusion.

6. PLOS authors have the option to publish the peer review history of their article (what does this mean?). If published, this will include your full peer review and any attached files.

Reviewer #1: **Yes: **Mahmoud AM Ali

Reviewer #2: **Yes: **Parvesh Mohan Garg

---

## [Author Response · Author response to Decision Letter 0]

9 Jul 2022

We deeply appreciate your valuable and kindest comments.

Reviewer 1. 

Thank you so much for your endorsement of our paper. 

Reviewer 2. 

Thank you for your precious comments for taking the time to point out options to improve our paper. We have revised the manuscript following your suggestions as follows. 

1. Please concise the previous studies related to meconium related ileus if possible in the introduction section.

: Thank you for your helpful comments. We’ve made concise the previous studies related to meconium related ileus in the introduction section. We’ve highlighted the sentences for you to check.

2. We define the primary aim and secondary aims more clearly in the hypothesis section.

: Thank you for your helpful comments. 

We’ve changed the sentence of aims in the background of the abstract and in the introduction of the main manuscript with your comments. We’ve highlighted the sentences for you to check. 

3. Can we combine the table 1 and 2. Keep the maternal factors first followed by infant factors.

: Thank you for your helpful comments. 

We’ve combined table 1 and 2 into table 1 with keeping the maternal factors first followed by infant factors. We’ve highlighted the sentences for you to check. 

4. Can we look into the factors why there is delay in start of the feeds after in gastrografin group?

: Thank you for your helpful comments. We added the reasons of the delay in the start of the feeding in discussion. We’ve highlighted the sentence for you to check. 

5. I would advised to keep the hypothesis focused to the characteristics of MRI and immediate outcomes.

6. I would take the growth outcomes out of the hypothesis.

: Thank you for your sincere comments (5.6). 

To the best of our knowledge, the current study is the first to report growth outcomes after resolved MRI by GaE treatment. We propose this is the merit of our study. Therefore, we are sorry that we cannot take out the growth outcome. I hope for your understanding and consideration. 

7. Can we change the MRI to meconium associated ileus. 

: Thank you for your sincere comments. In 1999, Kubota et al. proposed that the term meconium-related ileus (MRI) should include meconium plug syndrome and meconium disease because both diseases have speculated similar pathogenesis [6]. We propose MRI is the most appropriate diagnosis for enrolled patients in our study. Therefore, we are sorry that we cannot change the MRI to meconium associated ileus. I hope for your understanding and consideration. 

8. The BPD effect is more related to gestation age rather than MRI. I would not include that in the conclusion.

: Thank you for your helpful comments. We’ve deleted the BPD effect in conclusion of the abstract and the main manuscript.

---

## [Decision Letter · Decision Letter 1]

29 Jul 2022

Clinical and growth outcomes after meconium-related ileus improved with Gastrografin enema in very low birth weight infants.

PONE-D-22-07282R1

Dear Dr. Kim,

We’re pleased to inform you that your manuscript has been judged scientifically suitable for publication and will be formally accepted for publication once it meets all outstanding technical requirements.

Kind regards,

Prem Singh Shekhawat, MD

Academic Editor

PLOS ONE

Additional Editor Comments (optional):

Reviewers' comments:

Reviewer's Responses to Questions

**Comments to the Author**

1. If the authors have adequately addressed your comments raised in a previous round of review and you feel that this manuscript is now acceptable for publication, you may indicate that here to bypass the “Comments to the Author” section, enter your conflict of interest statement in the “Confidential to Editor” section, and submit your "Accept" recommendation.

Reviewer #1: All comments have been addressed

Reviewer #2: All comments have been addressed

2. Is the manuscript technically sound, and do the data support the conclusions?

Reviewer #1: Yes

Reviewer #2: Yes

3. Has the statistical analysis been performed appropriately and rigorously? 

Reviewer #1: Yes

Reviewer #2: Yes

4. Have the authors made all data underlying the findings in their manuscript fully available?

Reviewer #1: Yes

Reviewer #2: Yes

5. Is the manuscript presented in an intelligible fashion and written in standard English?

Reviewer #1: Yes

Reviewer #2: Yes

6. Review Comments to the Author

Reviewer #1: (No Response)

Reviewer #2: Thank you for addressing all the raised questions.

I would advise grammar check again before publication.I have no further comments and suggestions.

7. PLOS authors have the option to publish the peer review history of their article (what does this mean?). If published, this will include your full peer review and any attached files.

Reviewer #1: No

Reviewer #2: **Yes: **Parvesh Mohan garg

---

## [Editor Report · Acceptance letter]

3 Aug 2022

PONE-D-22-07282R1 

Clinical and growth outcomes after meconium-related ileus improved with Gastrografin enema in very low birth weight infants. 

Dear Dr. Kim:

I'm pleased to inform you that your manuscript has been deemed suitable for publication in PLOS ONE. Congratulations! Your manuscript is now with our production department. 

Kind regards, 

on behalf of

Dr. Prem Singh Shekhawat 

Academic Editor

PLOS ONE